# Suitability of GnRH Receptors for Targeted Photodynamic Therapy in Head and Neck Cancers

**DOI:** 10.3390/ijms20205027

**Published:** 2019-10-11

**Authors:** Lilla Pethő, József Murányi, Kinga Pénzes, Bianka Gurbi, Diána Brauswetter, Gábor Halmos, Gabriella Csík, Gábor Mező

**Affiliations:** 1MTA-ELTE Research Group of Peptide Chemistry, 1117 Budapest, Hungary; 2MTA-SE Pathobiochemistry Research Group, 1094 Budapest, Hungary; 3Department of Biopharmacy, Faculty of Pharmacy, University of Debrecen, 4032 Debrecen, Hungary; 4SE Department of Biophysics and Radiation Biology, 1094 Budapest, Hungary; 5Department of Organic Chemistry, Faculty of Science, Institute of Chemistry, ELTE Eötvös Loránd University, 1117 Budapest, Hungary

**Keywords:** GnRH-R, GnRH, protoporphyrin, targeted drug delivery, conjugate, photodynamic therapy, head and neck cancer

## Abstract

Head and neck squamous cell carcinomas (HNSCC) have a high mortality rate, although several potential therapeutic targets have already been identified. Gonadotropin-releasing hormone receptor (GnRH-R) expression is less studied in head and neck cancers, hence, we investigated the therapeutic relevance of GnRH-R targeting in HNSCC patients. Our results indicate that half of the patient-derived samples showed high GnRH-R expression, which was associated with worse prognosis, making this receptor a promising target for GnRH-based drug delivery. Photodynamic therapy is a clinically approved treatment for HNSCC, and the efficacy and selectivity may be enhanced by the covalent conjugation of the photosensitizer to a GnRH-R targeting peptide. Several native ligands, gonadotropin-releasing hormone (GnRH) isoforms, are known to target GnRH-R effectively. Therefore, different ^4^Lys(Bu) modified GnRH analogs were designed and conjugated to protoporphyrin IX. The receptor binding potency of the novel conjugates was measured on human pituitary and human prostate cancer cells, indicating only slightly lower GnRH-R affinity than the peptides. The *in vitro* cell viability inhibition was tested on Detroit-562 human pharyngeal carcinoma cells that express GnRH-R in high levels, and the results showed that all conjugates were more effective than the free protoporphyrin IX.

## 1. Introduction

Photodynamic therapy (PDT) is a clinically approved treatment for malignant tumors, such as head and neck, skin, breast, and bladder cancers. It combines nontoxic components as follows: A photosensitizer, light, and oxygen. The photosensitizing agent can passively accumulate into tumor cells, but it is nontoxic in the absence of light. If it is irradiated at a wavelength corresponding to an absorbance band of the sensitizer in the presence of molecular oxygen, a series of events lead to direct tumor cell death. Electron and energy transfers generate highly reactive radicals or singlet oxygen (^1^O_2_) that rapidly react with biomolecules leading to cell death via apoptosis or necrosis. Selectivity is derived from both the ability of photosensitizers to localize in the tumor and the precise delivery of light to the treated sites. Unfortunately, most commonly applied photosensitizers used for PDT accumulate not only in tumors, but also in healthy tissues, resulting in severe side effects like prolonged skin and eye photosensitivity [1,2,3,4]. Therefore, targeting studies in which the photosensitizers are covalently attached to various molecules (e.g., peptides) that have a specific affinity to receptors overexpressed on tumors are currently intensively investigated [5].

Gonadotropin-releasing hormone (GnRH) receptors are overexpressed on several tumor cells. These receptors are not only expressed by reproductive organ-derived tumors (ovarian, endometrial, prostatic, and breast cancers), but also oral and laryngeal cancers, renal carcinomas, brain tumors, melanomas, liver, pancreatic and colon carcinomas, and non-Hodgkin lymphomas [6]. One of the native ligands of the GnRH receptors (GnRH-R) found in humans is GnRH-I. This hormone peptide is a decapeptide (<EHWSYGLRPG-NH_2_, where <E is pyroglutamic acid) synthesized and released in the hypothalamus and plays a pivotal role in the control of vertebral reproduction by regulating gonadal activity and stimulating the secretion of pituitary luteinizing hormone (LH) and follicle stimulating hormone (FSH) [7]. GnRH-II (<EHWSHGWYPG-NH_2_) is another isoform, mainly expressed in the kidneys, bone marrow, and prostate [8], and is found to be a neuromodulator that stimulates sexual behavior [9]. GnRH-III (<EHWSHDWKPG-NH_2_) was originally isolated from lamprey. It binds to both type I and type II GnRH receptors [10], moreover it can inhibit proliferation of different cancer cells without significant hormonal activity [11].

The *N*- and *C*-terminal parts of GnRH (amino acids 1–4 and 9–10), which are essential for receptor binding and activation, are conserved [9], but the amino acids 5–8 can be changed without significant loss of efficacy. In the sequence of GnRH-I and GnRH-II, the glycine in position 6 can be replaced by *D*-lysine, which serves as conjugation site, increases enzymatic stability, and enhances the agonistic effect [12]. ^6^*D*-Lys in GnRH-I and GnRH-II analogs was used for daunorubicin (Dau) conjugation directly or through an enzyme-labile spacer with oxime linkage. It was demonstrated that the long-term cytotoxic effect of these conjugates was comparable with the free drug [13]. Schally and his coworkers developed AN-152, where the hemiglutarate derivative of doxorubicin was conjugated to the side chain of ^6^*D*-Lys in GnRH-I analog, resulting in an ester bond between the drug molecule and the peptide [14]. Extensive *in vitro* [15,16] and *in vivo* [17,18,19,20] studies were done and AN-152 reached clinical trials [21,22], but it failed in phase III since it could not improve overall survival, progression-free survival, objective response rate, clinical benefit rate, or adverse events compared to doxorubicin as a second line therapy for advanced endometrial cancers [23]. AN-152 was also used to target oral (KB) and laryngeal (HEp-2) carcinoma cells and was found to be very effective on both GnRH-R expressing tumors. In addition, it could overcome resistance to doxorubicin [24]. Rahimipour et al. have already conjugated protoporphyrin IX to GnRH-I [^6^*D*-Lys] and the selective receptor mediated phototoxicity could be demonstrated on a αT3-1 pituitary gonadotrope cell line [25]. In another study, carminic acid, a hydroxyanthraquinone derivative, was conjugated to GnRH-I [^6^*D*-Lys] to gain a phototoxic molecule and retained bioactivity, and formation of reactive oxygen species upon irradiation was observed [26].

In the case of GnRH-III, the natural sequence contains two side chain functional groups (^6^Asp and ^8^Lys) that can serve as conjugation sites. Previous studies showed that modification of the side chain of ^8^Lys did not change the antiproliferative activity or the receptor binding [27] and, furthermore, it reduced the endocrine effect [28]. Hence, this position can be used for further conjugations. It was also indicated that the replacement of ^4^Ser to Lys or Lys(Ac) results in similar antitumor activity and endocrine effect [29]. Moreover, the modification of the ^4^Lys side chain with butyric acid (Lys(Bu)), a short-chain fatty acid, significantly increased the stability against chymotrypsin and the *in vitro* cellular uptake of oxime bond-linked Dau-GnRH-III bioconjugates [30] and led to enhanced *in vivo* antitumor activity [31]. These results presume that the same modification (^4^Ser → ^4^Lys(Bu)) in the sequence of GnRH-I and GnRH-II can result in similar advantageous effects.

Protoporphyrin IX (PpIX) is an endogenous photosensitizer and it is the last intermediate in heme biosynthesis. Endogenous PpIX-based strategies have been approved by the FDA for treating cancer, where δ-aminolevulinic acid (ALA, the first intermediate in heme biosynthesis) is administered orally or locally to generate PpIX biosynthesis. Unfortunately, the generated PpIX does not only accumulate in cancer cells but also in healthy cells, such as the marrow, the circulating erythrocytes, and the liver, causing photosensitivity or liver damage [32]. PpIX has two carboxyl groups that are suitable for the conjugation of a targeting moiety, giving the opportunity to increase the selectivity. Hence recently, PpIX has also been studied as exogenous photosensitizer conjugated to peptides [25,33], nanoparticles [34,35,36], or quantum dots [37] and encapsulated into polymer dendrimers [38,39]. In this study, PpIX was conjugated to the novel ^4^Lys(Bu) modified GnRH-analogs to overcome undesirable side-effects by enhancing the selectivity and efficacy of the treatment. Our aim was to compare the effectiveness of the different GnRH conjugates and to prepare more effective compounds than PpIX. PDT can be used in those types of cancer that are easily accessible for the irradiation, such as head and neck cancers, melanomas, or lung cancers. Hence, in the present study, GnRH receptor expression was investigated in patent-derived head and neck squamous cell carcinoma (HNSCC) samples by immunohistochemistry. Based on the positive results, our novel bioconjugates were tested *in vitro* on Detroit-562 human pharyngeal carcinoma cells that have already been demonstrated to express GnRH receptors in high levels [40].

## 2. Results

### 2.1. GnRH receptor (GnRH-R) Expression in Patient-Derived Head and Neck Squamous Cell Carcinoma (HNSCC) Samples

From the 60 tumor samples, 8 (13.3%) cases showed low, 25 (41.7%) cases showed moderate, and 27 (45.0%) cases showed high GnRH-R expression (Figure 1 and Figure 2A). For statistical analysis, scores were dichotomized along different threshold values. The most reproducible threshold for the assessor was set up when scores of 1 and 2 were considered low protein expression, whereas scores of 3 were taken as high protein expression.

The GnRH-R status did not correlate with tumor size (*p* = 0.722), tumor localization (*p* = 0.527), lymph node metastasis (*p* = 0.126), stage (*p* = 0.913), and disease-specific survival (DSS, *p* = 0.423). However, the increase in GnRH-R expression was associated with worse prognosis (Figure 2B).

### 2.2. Peptide Synthesis

GnRH-I and GnRH-II analogs (<EHWSYkLRPG-NH_2_, **1**; <EHWK(Bu)YkLRPG-NH_2_, **2**; and <EHWK(Bu)HkWYPG-NH_2_, **3**) were prepared using *D*-Lys in position 6. Methyl ester protected aspartic acid was used in position 6 in case of the GnRH-III analog (<EHWK(Bu)HD(OMe)WKPG-NH_2_, **4**) to ensure that protoporphyrin IX (PpIX) can couple only to ^8^Lys. It was shown that this protecting group did not influence the efficacy of GnRH-III significantly [41]. For the synthesis of the butyric acid modified analogs, selectively protected lysine derivative was incorporated in position 4. After the synthesis of the backbones, this selectively removable protecting group was cleaved and the free ε-amino group of the lysine was reacted with butyric anhydride.

### 2.3. GnRH-Protoporphyrin IX Conjugate Synthesis

The amide bond formation between the ε-amino groups of the appropriate lysine residues and one of the carboxyl groups of PpIX was carried out in solution (Figure 3), resulting in compounds **5**–**8** (Table 1). Since protoporphyrin is a photolabile compound, these reactions were performed in darkness. Only 1:1 peptide-PpIX conjugates were formed (no dimer conjugates—2:1 peptide-PpIX— could be observed) and only a small amount of water-lost side-product was detected in case of the GnRH-I analogs, which could be hydrolyzed under basic conditions to the desired product, presuming an extra ester bond formation between the other carboxyl group of PpIX and the tyrosine side chain. This side product was not detected in the case of GnRH conjugates without Tyr content.

### 2.4. Receptor Binding Potency of the Novel Compounds

The binding potency of the GnRH analogs and the GnRH-PpIX conjugates to human pituitary and human prostate cancer cells, that express GnRH-R in high concentration, was investigated by ligand competition assay. The displacement of ^125^I-radiolabelled GnRH-I agonist triptorelin ([^125^I]-[^6^*D*-Trp]-GnRH-I) by the synthesized compounds was determined. At a concentration of 10^−6^ M unconjugated PpIX and several other peptides structurally related or structurally and/or functionally unrelated to GnRH, such as somatostatin-14, hGHRH, EGF, IGF-I, glucagon and VIP did not inhibit the binding of [^125^I]-[^6^*D*-Trp]-GnRH-I. The data showed (Table 2) that the ^4^Lys(Bu) modified GnRH analogs (**2**–**4**) could bind GnRH-R with similar potency than GnRH-I[6*D*-Lys] (**1**), while the PpIX conjugation decreased the receptor binding affinity in all cases. Nevertheless, the peptide-PpIX conjugates (**5**–**7**) had only slightly lower GnRH-R affinity, except compound **8** that had two orders of magnitude lower binding affinity to the receptors than the free peptide.

### 2.5. UV-Vis Absorbance

The UV-Vis spectra of the conjugates were measured in PBS-acetonitrile mixture to determine any changes in the absorbance compared to protoporphyrin. There was no significant change observed in the spectra (Figure 4), indicating that the conjugation did not change the features of PpIX. (As demonstrated in Appendix A, the peptides alone did not have a significant contribution to the absorbance above 300 nm in the UV spectra.)

### 2.6. GnRH-R Expression of Detroit-562 Human Pharyngeal Carcinoma Cells

Detroit-562 pharynx cells were selected for the *in vitro* biological studies of the GnRH-PpIX conjugates. This cell line has already been proved to express GnRH-R [40] and our measurements, using confocal laser scanning microscopy, also demonstrated high expression level of GnRH-R (Figure 5, GnRH-R in permeabilized cells). A significant number of receptors was found in non-permeabilized cells (Figure 5, GnRH-R in non-permeabilized cells), as well, which may refer to the presence of GnRH-R in the cell membrane.

### 2.7. Optimization of the Irradiation and Incubation Time

GnRH-I[^6^*D*-Lys(PpIX)] (compound **5**) was used for the optimization of the conditions of the cell viability inhibition assays. First, the irradiation time was optimized. Detroit-562 pharyngeal cells were treated with the peptide-PpIX conjugate for 5 h, the unbound conjugate was washed out, and then the cells were irradiated for 0, 2, 10, or 30 min followed by 72 h of further incubation (Figure 6A). The obtained results showed that the conjugate was non-toxic in the absence of light, while irradiation for 2 min resulted in 50% cell viability inhibition in sub-micromolar concentration. There was no significant difference between the 10 and the 30 min irradiation; both were already very effective in 0.1 µM concentration (~98% cell viability inhibition), therefore 10 min irradiation was chosen for the further experiments. Moreover, this length of time may also be more favorable in the subsequent *in vivo* studies.

Afterwards, the incubation time was optimized. The cells were incubated either with compound **5** or with PpIX for 1, 3, or 5 h and irradiated for 10 min. As demonstrated in Figure 6B, a longer incubation time with compound **5** resulted in higher activity. On the contrary, there was no significant difference between the effect of the incubation times in case of the protoporphyrin (Figure 6C). Based on these results, the 5 h incubation time was chosen for the further measurements.

### 2.8. In Vitro Cell Viability Inhibition Effect of the Peptide-PpIX Conjugates

The cell viability inhibition of the synthesized GnRH-PpIX conjugates was measured on Detroit-562 cells using the previously optimized conditions (5 h treatment, 10 min irradiation). Figure 7 shows that all conjugates were more active than PpIX. All were very effective (90–95% inhibition) in 0.25 µM, while PpIX showed only ~50% inhibition at this concentration. Conjugates **5** and **7** were the most active compounds. These showed >90% cell viability inhibition at 0.1 µM. Conjugates **6** and **8** were less effective, as both analogs showed 65–70% inhibition at 0.1 µM concentration. These results are also confirmed by the calculated IC_50_ values (Table 3). The peptides (**1**–**4**) alone did not show cell viability inhibition, even at 10 µM concentration (Appendix A).

## 3. Discussion

Head and neck squamous cell carcinomas (HNSCC) account for ~4% of all malignancies worldwide. HNSCC has a high mortality rate and is more common among men and those over the age of 50 [42]. Several potential therapeutic targets have already been identified (e.g., epidermal growth factor receptor (EGFR)) [43,44], however, targeted therapy of HNSCC is still not well established due to acquired resistance [45]. Hence, finding other appropriate molecular targets is essential to achieve personalized targeted therapy. GnRH receptor (GnRH-R) presence has already been proved by displacement binding assays on KB oral and HEp-2 laryngeal carcinoma cells [24], while high GnRH-R expression and cell surface localization, moreover effective cellular uptake of GnRH analogs, has been demonstrated on Detroit-562 human pharynx tumor cells [40]. Since no comprehensive research has been done so far about GnRH-R in HNSCC, we investigated 60 patient-derived tumor samples and tested them for GnRH-R. The results indicated that 45.0% of these tumors showed high GnRH-R expression, which makes this receptor a promising target in the treatment of HNSCC. Moreover, we checked several clinicopathological factors in correlation of GnRH-R status and we found that higher GnRH-R expression was associated with worse prognosis. We also strengthened that the Detroit-562 human pharyngeal carcinoma cells express GnRH-R in high levels and a high proportion of receptors is found on non-permeabilized cells as well, implying that these receptors may be localized in the cell membrane, proving that these cells are appropriate for GnRH-R targeted *in vitro* studies.

PDT is a clinically approved treatment for HNSCC since all types (oropharynx, larynx, hypopharynx) are easily accessible for irradiation. The covalent conjugation of a photosensitizer to a proper delivery molecule that targets GnRH-R may enhance the efficacy and selectivity of PDT. To find the most effective targeting moiety we designed three novel butyric acid modified GnRH derivatives (^4^Lys(Bu) modification) and compared them to the generally used GnRH-I superagonist GnRH-I[^6^*D*-Lys]. The receptor binding potency of the novel peptides (**2**–**4**) was similar to the control peptide (**1**), presuming that all analogs have similar targeting capability. Then, PpIX as photosensitizer was coupled to the GnRH-analogs via amide bond under controlled conditions to get the well-defined 1:1 conjugates solely. We demonstrated that the conjugation of PpIX to the peptides did not affect the UV-Vis absorbance of the photosensitizer. These GnRH-PpIX conjugates showed lower receptor binding affinity than the free peptides, probably due to steric hindrance, however, this was only a slight difference in case of compounds **5**–**7**. Conjugate **8** showed two orders of magnitude difference compared to **4**. It is worth mentioning that GnRH-I and GnRH-II have U shape structures and the photosensitizer attached to the *D*-Lys in position 6 cannot interfere with the termini of the peptides that take part in the receptor recognition. In contrast, GnRH-III shows, rather, a flexible extended structure. Therefore, the attachment of a bulky compound to Lys in position 8 might significantly decrease the binding potency of the conjugate to the receptor [46].

After the optimization of the *in vitro* cell viability assay conditions (5 h of incubation time and 10 min of irradiation) we tested all GnRH-PpIX conjugates on Detroit-562 cells. We could demonstrate that all synthesized conjugates were more effective than the free PpIX. Compounds **5** and **7** were the most active conjugates. These showed an outstanding cell viability inhibition effect already at 0.1 µM, while **6** and **8** were less effective. All conjugates showed >90% cell viability inhibition at 0.25 µM; however, PpIX was not very effective in sub-micromolar concentration. These results correspond well with the receptor binding affinity of the conjugates. According to the results, the ^4^Lys(Bu) modified GnRH-II analog showed similar efficacy to the control GnRH-I analog, while the modified GnRH-I analog and the GnRH-III derivative were just slightly less effective.

In this work, we demonstrated that GnRH-R may be an appropriate target in the treatment of HNSCC and, to prove this statement, four different GnRH-PpIX conjugates were synthesized for PDT. The *in vitro* cell viability inhibition was studied on a pharyngeal cell line and our results confirmed that these conjugates may be used later in PDT of HNSCC. All conjugates were more effective than the free PpIX, showing the importance and advantage of targeted therapy in PDT. Based on these results, we plan to test the two best conjugates in *in vivo* models as well.

## 4. Materials and Methods

### 4.1. Chemicals

All amino acid derivatives, the Fmoc-Rink Amide MBHA resin, *N*,*N*′-diisopropylcarbodiimide (DIC), and trifluoroacetic acid were purchased from Iris Biotech GmBH (Marktredwitz, Germany). The 1-hydroxybenzotriazole hydrate (HOBt), triisopropylsilane (TIS), diisopropylethylamine (DIPEA), butyric anhydride, hydrazine hydrate, and protoporphyrin IX (PpIX) were obtained from Sigma Aldrich Kft. (Budapest, Hungary). The 1,8-diazabicyclo[5.4.0]undec-7-ene (DBU) was from TCI Europe N.V. (Zwijndrecht, Belgium). Benzotriazol-1-yl-oxytripyrrolidinophosphonium hexafluorophosphate (PyBOP) was obtained from Bachem AG (Bubendorf, Switzerland), while piperidine was purchased from Molar Chemicals Kft (Budapest, Hungary). All solvents used for synthesis and purification were purchased from VWR International Kft. (Debrecen, Hungary). All reagents and solvents were of analytical grade or the highest available purity.

For the *in vitro* biological assays, EMEM (Eagle’s minimum essential medium), sodium pyruvate, MycoZap, and PBS were obtained from Lonza Group Ltd. (Basel, Switzerland), FBS and trypsin were purchased from Gibco^®^/Invitrogen Corporation (New York, NY, USA), BSA, and mounting medium was from Millipore Sigma (Burlington, MA, USA). Draq5™and DAB Quanto were purchased from Thermo Fisher Scientific (Waltham, MA, USA), while the MTT powder was obtained from Sigma Aldrich Kft. (Budapest, Hungary). *h*GnRH-I-R primary antibody and GnRH-R polyclonal antibody were purchased from Proteintech™ (Rosemont, IL, USA) and Alexa Fluor 594 conjugated secondary antibody was from Jackson ImmunoResearch Inc. (Ely, UK).

### 4.2. Patients

Altogether, 60 therapy naive patients, who were diagnosed with squamous cell carcinoma of the oropharynx, hypopharynx, and larynx at the Department of Oto-Rhino-Laryngology and Head and Neck Surgery, Semmelweis University between 2012 and 2014, were consecutively enrolled. All subjects gave their informed consent for inclusion before they participated in the study. The study was conducted in accordance with the Declaration of Helsinki and the protocol was approved by the Semmelweis University’s Regional, Institutional Scientific, and Research Ethics Committee (ethical license No: 105/2014). The most important characteristics of our cohort are shown in Table 4.

### 4.3. Tissue Microarray (TMA) and Immunohistochemistry

TMA blocks containing 2 mm diameter cores of formalin-fixed paraffin-embedded tissue samples were created using the TMA Master tool (3DHISTECH Kft, Budapest, Hungary). Tissue sections (4 μm) were cut on adhesion slides and were stained with hematoxylin and eosin and GnRH-R. The protocol of the staining method was carried out as described previously [47].

Immunohistochemistry was performed in TMA sections following routine dewaxing rehydration. For antigen retrieval, the samples were boiled in Tris-EDTA buffer solution (pH 9.0) for 58 min. Endogenous peroxidase activity was blocked using 3% H_2_O_2_ in methanol for 15 min. Immunostaining included the following sequential incubation steps: Usage of 3% BSA in 0.1 M Tris-buffered saline with Tween^®^ 20 pH 7.4 (TBST) as a protein block for 15 min, the optimally diluted primary antibody (1:100) overnight (16 h), and the HISTOLS-MR-T HRP Polymer reagent for 40 min. Samples were washed after each incubation step for 10 min in TBST. Peroxidase activity was visualized using DAB Quanto for 1 min under microscopic control. Finally, nuclear counterstaining was applied using hematoxylin and eosin. All incubations were performed in humidity chambers at room temperature. Immunostained slides were digitalized applying a Pannoramic Scan instrument (3DHISTECH Kft, Budapest, Hungary). The histologic evaluation and the scoring of immunoreactions were done by 2 independent assessors using the Pannoramic Viewer software.

An alternative 3-grade scoring approach, applied in several earlier studies for the evaluation of EGFR protein expression [48,49], was used for the evaluation of GnRH-R protein expression. Briefly, the percentage of stained cells was multiplied by the grade intensity of staining (in four grades: 1, negative; 2, weak; 3, moderate; 4, intense), which resulted in a value between 0 and 400. Cores with scores 0 to 200 (1), 201 to 300 (2), and 301 to 400 (3) were referred to as weak, medium, or strong protein expression, respectively. For statistical analysis, scores were dichotomized along different threshold values. The most reproducible threshold for the assessor was set up when scores of 1 and 2 were considered low/negative protein expression, whereas scores of 3 were taken high/positive protein expression.

### 4.4. Statistical Analysis

For patient data, statistical analysis was performed using Statistica 13 (TIBCO Software Inc., Palo Alto, CA, USA). The Pearson *χ*^2^ tests and the Fisher’s exact tests were used to test correlations between discrete variables. In case of survival analysis, Kaplan–Meier estimation with log-rank test as well as univariate and multivariate regression were applied. All tests were two-sided and *p*-values <0.05 were considered statistically significant. Tumor localization, tumor size, stage, lymph node metastasis, and the biomarkers listed above were used in the analysis.

### 4.5. Peptide Synthesis

All peptides were synthesized manually using solid phase peptide synthesis according to standard Fmoc/*t*Bu strategy. The Fmoc deprotection was performed with 2% piperidine + 2% DBU in DMF (2 + 2 + 5 + 10 min), then the α-Fmoc-protected amino acid derivatives were coupled with DIC and HOBt (3 eq each to the resin capacity).

#### 4.5.1. GnRH-I[^6^*D*-Lys] (Compound **1**)

The GnRH-I modified with *D*-Lys in position 6 (<EHWSYkLRPG-NH_2_) was synthesized on Fmoc-Rink Amide MBHA resin (0.69 mmol/g coupling capacity). After the completion of the synthesis, the decapeptide was cleaved from the resin using a mixture of 95% TFA, 2.5% H_2_O, and 2.5% TIS for 1.5 h at room temperature. The crude peptide was precipitated into ice-cold diethyl ether, centrifuged, and washed three times with diethyl ether. The precipitate was dissolved in a mixture of acetonitrile and water prior lyophilization, then the peptide was purified by preparative RP-HPLC and analyzed by ESI-MS.

#### 4.5.2. GnRH-I[^4^Lys(Bu),^6^*D*-Lys] (Compound **2**) and GnRH-II[^4^Lys(Bu),^6^D-Lys] (Compound **3**)

Analogs modified with butyrylated lysine (Lys(Bu)) in position 4 and with *D*-Lys in position 6 (GnRH-I: <EHWK(Bu)YkLRPG-NH_2_ and GnRH-II: <EHWK(Bu)HkWYPG-NH_2_) were synthesized similarly as described above. In these cases, Fmoc-Lys(Dde)-OH was incorporated into the peptide in position 4. After completion of the protected decapeptides, the Dde group of ^4^Lys was removed on the resin by 2% N_2_H_4_·H_2_O in DMF (8 × 5 min), then the acylation was performed using butyric anhydride and DIPEA (3 eq each to the amino group) in DMF for 2 h. The peptides were cleaved from the resin and analyzed as described above.

#### 4.5.3. GnRH-III[^4^Lys(Bu),^6^Asp(OMe)] (Compound **4**)

GnRH-III analog modified with Lys(Bu) in position 4 and with aspartic acid methyl ester (Asp(OMe)) in position 6 (<EHWK(Bu)HD(OMe)WKPG-NH_2_) was synthesized similarly as described above, but since the methyl ester group is labile under basic conditions, Fmoc-Lys(Mtt)-OH was used in position 4. Furthermore, the mixture used for Fmoc deprotection was changed to 0.1 M HOBt + 2% piperidine + 2% DBU in DMF to avoid the methyl ester cleavage and the subsequent succinimide bond formation. After the completion of the backbone synthesis, the Mtt group was removed from the ε-amino group by 2% TFA + 2% TIS in DCM (2 + 2 + 5 + 10 + 30 min) and the lysine was butyrylated (with butyric anhydride and DIPEA; 3 eq each to the amino group). The peptide was cleaved from the resin and analyzed as described above.

### 4.6. Conjugate Synthesis (Compounds **5**–**8**)

The peptides, protoporphyrin IX and PyBOP (1:1:3 eq), were dissolved in DMF, then DIPEA (6 eq) was added to the mixture. The reaction was stirred for 16 h at room temperature in darkness. The bioconjugates were purified by preparative RP-HPLC and the pure conjugates were analyzed by analytical RP-HPLC and ESI-MS (analytical data, HPLC chromatograms and MS spectra of the produced conjugates are presented in Appendix A)

### 4.7. RP-HPLC

The crude peptides and the bioconjugates were purified on a Knauer 2501 HPLC system (H. Knauer, Bad Homburg, Germany). A preparative Phenomenex Luna C18 column (250 × 21.2 mm) with 10 µm silica (100 Å pore size) was used for the crude peptides, while a Phenomenex Jupiter C4 column (250 × 10 mm) with 10 µm silica (300 Å pore size) was used for the bioconjugates (Torrance, CA, USA). Linear gradient elution (0 min 5% B; 5 min 5% B; 50 min 80% B) with eluent A (0.1% TFA in water) and eluent B (0.1% TFA in acetonitrile-water 80:20, *v/v*) was used at a flow rate of 16 mL/min for the peptides and linear gradient elution (0 min 15% B; 5 min 15% B; 50 min 60% B) with eluent A (0.1% TFA in water) and eluent B (0.1% TFA in acetonitrile) was used at a flow rate of 4 mL/min for the bioconjugates. Peaks were detected at 220 nm.

Analytical RP-HPLC runs were performed on a Knauer 2501 HPLC system using a Macherey–Nagel Nucleosil C18 column (250 × 4.6 mm) with 5 µm silica (100 Å pore size). Linear gradient elution (0 min 2% B; 5 min 2% B; 30 min 90% B) with eluent A (0.1% TFA in water) and eluent B (0.1% TFA in acetonitrile) was used at a flow rate of 1 mL/min. Peaks were detected at 220 nm.

### 4.8. Mass Spectrometry

Electrospray (ESI)-mass spectrometric measurements were performed on an Esquire 3000+ ion trap mass spectrometer (Bruker Daltonics, Bremen, Germany). Spectra were acquired in positive mode in the 50–2000 *m/z* range. Samples were dissolved in a 0.1% acetic acid containing acetonitrile–water (50:50, *v/v*) mixture.

### 4.9. Radioligand Binding Assay

Displacement studies using ^125^I-labelled GnRH-I agonist triptorelin for the determination of the receptor binding potency of the synthesized compounds were performed on human pituitary and human prostate cancer cells. Tissue samples derived by autopsy from normal human pituitary (anterior lobe) and human prostate cancer cells were obtained from a patient at the time of initial surgical treatment. The collection and the use of these specimens for our studies are approved by the local Institutional Ethics Committee. The binding affinities of the GnRH-analogs and GnRH-protoporphyrin conjugates to GnRH-R-I were determined by displacement of [^125^I]-[^6^*D*-Trp]-GnRH-I performing an *in vitro* ligand competition assay. The assay was performed as described earlier [13,14,15,30,41,50]. In brief, membrane homogenates containing 50–160 µg protein were incubated in duplicate with 60–80,000 cpm [^125^I]-[^6^*D*-Trp]-GnRH-I in the presence of increasing concentrations (10^−12^–10^−6^ M) of nonradioactive unlabeled peptides/conjugates as competitors in a total volume of 150 µL of binding buffer. At the end of the incubation, 125 µL aliquots of suspension were transferred onto the top of 1 mL of ice-cold binding buffer containing 1.5% BSA in microcentrifuge tubes. The tubes were centrifuged at 12,000× *g* for 3 min at 4 °C. Supernatants were aspirated and the pellet was counted in a gamma counter. Protein concentration was determined by the method of Bradford, using a Bio-Rad protein assay kit (Bio-Rad Laboratories, Hercules, CA, USA). The LIGAND-PC computerized curve-fitting program of Munson and Rodbard was used to determine the receptor binding characteristics and IC_50_ values.

### 4.10. UV-Vis Spectroscopy

UV-Vis spectra were recorded between 200–800 nm using a BioTek Synergy 2 Multi-Mode Reader. The 50 μM solutions were prepared in PBS:acetonitrile (1:1, *v/v*) containing 1% DMSO.

### 4.11. Cell Cultures

Detroit-562 human pharyngeal cells were purchased from the American Type Culture Collection (Manassas, VA, USA). Cells were cultured in EMEM medium, containing 10% FBS, 1% sodium pyruvate, and 1% MycoZap, at 37 °C in a humidified 5% CO_2_ atmosphere. For the experiments, Detroit-562 cells with passage number 12 were thawed and serially passaged until passage 35. For new passages, cells were seeded into T25 flasks and grown to 80–90% confluence. Then the cells were rinsed with phosphate buffered saline and removed by adding Trypsin–EDTA. The cell number was established using a hemocytometer.

### 4.12. Confocal Laser Scanning Microscopy

Cells were seeded on Ibidi^®^ μ-Slide 8-well microscopic slides (10^4^ cells/well) 48 h prior to the treatment. Cells were fixed with 4% paraformaldehyde for 10 min and washed twice with PBS. In contrast of the membrane GnRH-R investigation, cells were also permeabilized with 0.1% (*v/v*) Triton X-100 PBS solution for 10 min in case of the total GnRH-R protein level determination. After blocking with 5% BSA in PBS for 30 min, the cells were incubated with *h*GnRH-I-R primary antibody (dilution: 1:100) and subsequently with Alexa Fluor 594 conjugated secondary antibody (dilution: 1:300) for 1-1 h at 25 °C. Then, Draq5™ fluorescent probe solution (10 µM) was added to the cells for 10 min. Cells were washed three times with PBS and a few drops of mounting media was added. Images of cells were acquired with confocal laser microscope (Zeiss Confocal LSM 710, Carl Zeiss AG, Oberkochen, Germany; objective: Plan-Apochromat 63x/1.40 Oil DIC M27; pinhole: 1 AU; laser wavelength: 543 nm 10.0%, 633 nm 4.0%; detection wavelength: 602–631 nm, 676–758 nm).

### 4.13. In Vitro Cell Viability Assay

Cells were seeded to 96-well plates (6·× 10^3^ cells/well) 48 h prior to the treatment. The cells were treated with the peptides (10 μM) and conjugates (0.025–10 μM) in darkness. Stock solutions of the compounds were prepared in DMSO (5 mM) and all dilutions were made to contain 0.2% DMSO final concentration. The control wells were treated with medium or 0.2% DMSO containing medium. After 1, 3, or 5 h, the cells were washed once with medium, irradiated for 0, 2, 10, or 30 min and incubated for a further 72 h. The irradiation was performed by an Oriel type 250 W Quartz Tungsten Halogen lamp (Newport Corporation, CA, USA) with a 550 nm emission maximum. The light was filtered to exclude UV and IR components. The fluence rate was determined by a 10 A-P Nova laser power/energy monitor (Ophir, Optronix, Jerusalem, Israel) using the pyro-electric detector. Then, the MTT-test was performed according to the manufacturer’s instructions. Each point was measured using 3 parallels and each experiment was repeated twice.

### 4.14. Statistical Evaluation of In Vitro Viability Data

For in vitro cell viability data, statistical analysis was performed using the multiple t-test algorithm of GraphPad Prism 8.2.0 software (GraphPad Software Inc., San Diego, CA, USA). Statistical significance was determined using the Holm–Sidak method. Each row was analyzed individually, without assuming a consistent SD. Significance levels correspond to * *p* < 0.05; ** *p* < 0.01; and *** *p* < 0.001.

## Figures and Tables

**Figure 1 ijms-20-05027-f001:**
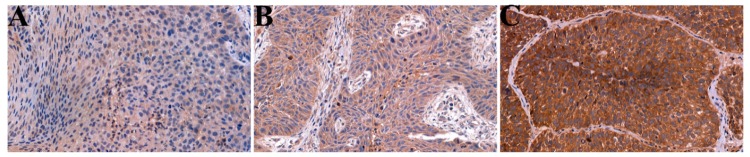
Examples of immunostaining in head and neck squamous cell carcinomas (HNSCC). (**A**) Low GnRH-R expression; (**B**) moderate GnRH-R expression; (**C**) high GnRH-R expression. (Magnification: ×40).

**Figure 2 ijms-20-05027-f002:**
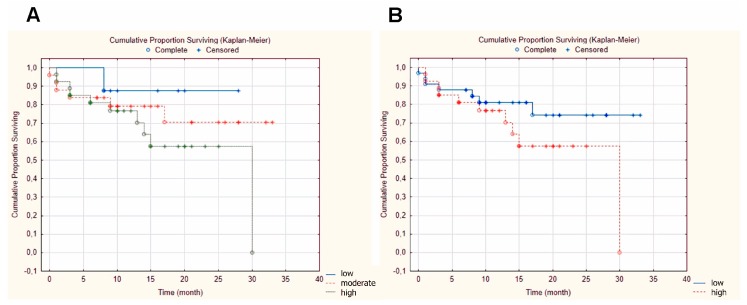
Kaplan–Meier survival curves. (**A**) Correlation between GnRH-R expression and disease-specific survival showing the 3 scores groups—low, moderate, and high expression (*p* = 0.556); (**B**) Correlation between GnRH-R expression and disease-specific survival showing the dichotomized scores groups—low and high expression (*p* = 0.423).

**Figure 3 ijms-20-05027-f003:**
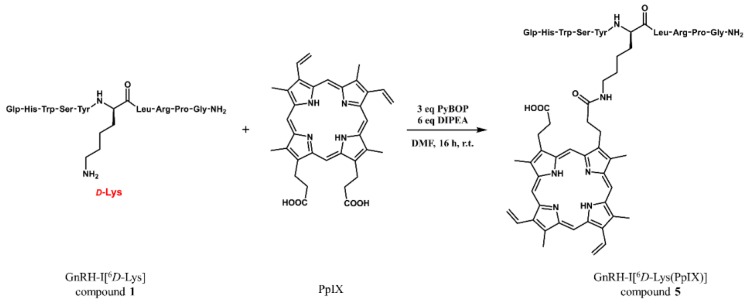
Synthesis of the GnRH-protoporphyrin conjugates on the example of GnRH-I[^6^*D*-Lys] (compound **1**) and PpIX reaction (resulting in compound **5**, GnRH-I[^6^*D*-Lys(PpIX)]). The coupling was performed in DMF with PyBOP (benzotriazol-1-yl-oxytripyrrolidinophosphonium hexafluorophosphate) in the presence of DIPEA (diisopropylethylamine) (16 h, r.t.).

**Figure 4 ijms-20-05027-f004:**
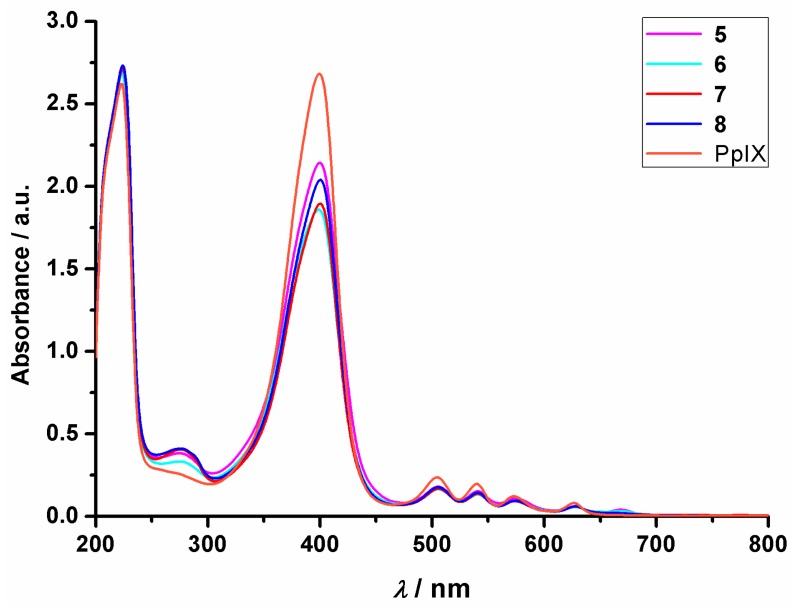
UV-Vis spectra of PpIX and the peptide-PpIX conjugates.

**Figure 5 ijms-20-05027-f005:**
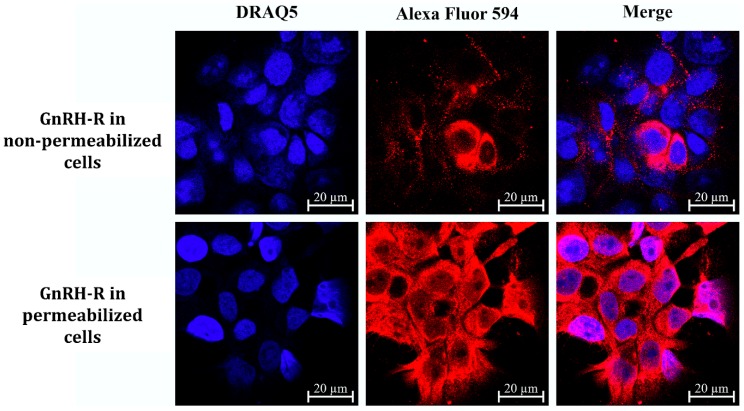
GnRH-R in Detroit-562 cells. The measurements confirm the high GnRH-R expression of Detroit-562 cells (GnRH-R in permeabilized cells) and the high number of receptors on non-permeabilized cells (nuclei: blue–DRAQ5; GnRH-R: red–Alexa Fluor 594).

**Figure 6 ijms-20-05027-f006:**
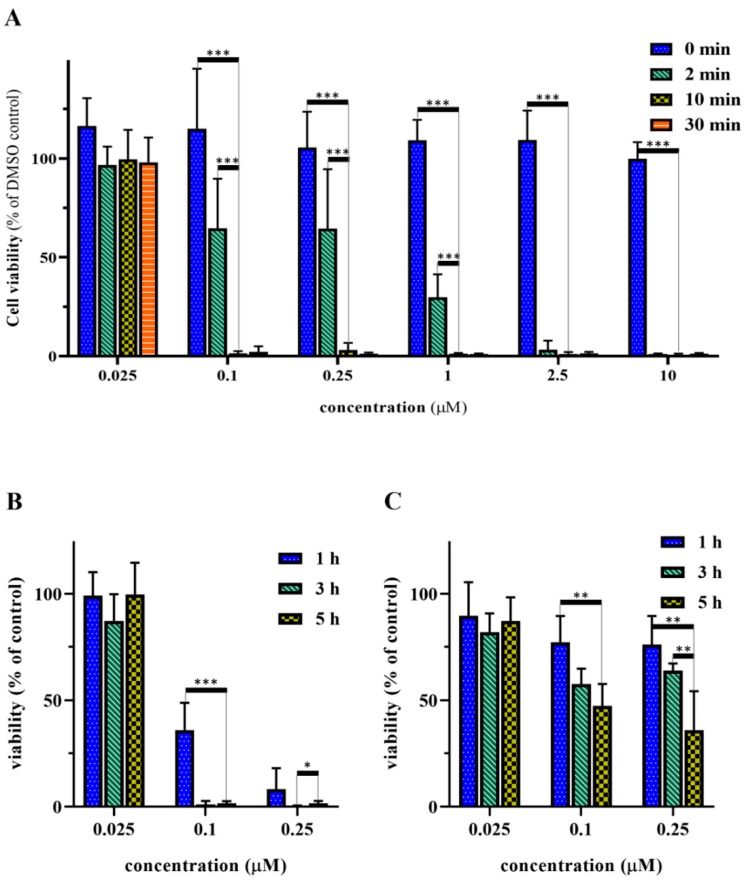
Optimization of the *in vitro* cell viability inhibition assay. (**A**) Effect of different irradiation times using compound **5**. (**B**) Effect of different incubation times using compound **5**. (**C**) Effect of different incubation times using PpIX. Cell viability values are present in % of DMSO control. Statistical significance was determined for the optimized values, i.e., 10 min irradiation time and 5 h incubation time (* *p* < 0.05; ** *p* < 0.01; *** *p* < 0.001).

**Figure 7 ijms-20-05027-f007:**
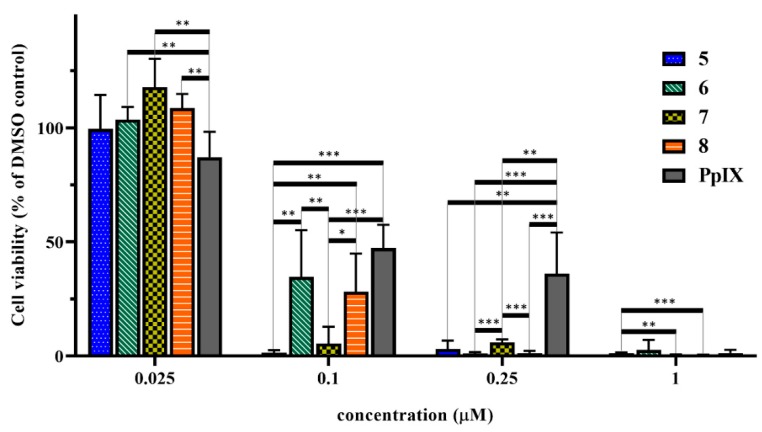
*In vitro* cell viability inhibition effect of the prepared GnRH-PpIX conjugates (compounds **5**–**8**) and PpIX using 5 h incubation and a 10 min irradiation period. Cell viability values are present in % of DMSO control. (* *p* < 0.05; ** *p* < 0.01; *** *p* < 0.001).

**Table 1 ijms-20-05027-t001:** The synthesized compounds: the linkable targeting peptides (GnRH analogs; **1**–**4**) and their PpIX conjugated derivatives (**5**–**8**).

Code	Peptide	Code	Conjugate
1	GnRH-I[^6^*D*-Lys]	5	GnRH-I[^6^*D*-Lys(PpIX)]
2	GnRH-I[^4^Lys(Bu), ^6^*D*-Lys]	6	GnRH-I[^4^Lys(Bu), ^6^*D*-Lys(PpIX)]
3	GnRH-II[^4^Lys(Bu), ^6^*D*-Lys]	7	GnRH-II[^4^Lys(Bu), ^6^*D*-Lys(PpIX)]
4	GnRH-III[^4^Lys(Bu), ^6^Asp(OMe)]	8	GnRH-III[^4^Lys(Bu), ^6^Asp(OMe), ^8^Lys(PpIX)]

**Table 2 ijms-20-05027-t002:** Receptor binding potency of the GnRH analogs and the GnRH-PpIX conjugates measured by the inhibition of [^125^I]-[^6^*D*-Trp]-GnRH-I binding to the membranes of human pituitary and human prostate cancer cells.

Code	Compound	IC_50_ Values/nM	
		Human Pituitary	Human Prostate Cancer
1	GnRH-I[^6^*D*-Lys]	6.44 ± 0.95	4.31 ± 0.83
2	GnRH-I[^4^Lys(Bu), ^6^*D*-Lys]	9.70 ± 1.07	8.61 ± 1.08
3	GnRH-II[^4^Lys(Bu), ^6^*D*-Lys]	5.56 ± 1.07	4.65 ± 0.31
4	GnRH-III[^4^Lys(Bu), ^6^Asp(OMe)]	7.27 ± 0.92	6.56 ± 1.11
5	GnRH-I[^6^*D*-Lys(PpIX)]	36.29 ± 3.17	42.67 ± 7.04
6	GnRH-I[^4^Lys(Bu), ^6^*D*-Lys(PpIX)]	42.30 ± 4.27	41.80 ± 5.74
7	GnRH-II[^4^Lys(Bu), ^6^*D*-Lys(PpIX)]	79.40 ± 8.88	84.50 ± 10.30
8	GnRH-III[^4^Lys(Bu), ^6^Asp(OMe), ^8^Lys(PpIX)]	288.6 ± 17.3	341.4 ± 21.1

**Table 3 ijms-20-05027-t003:** Calculated IC_50_ values of the prepared conjugates and PpIX.

Code	Compound	IC_50_ Values/nM
5	GnRH-I[^6^*D*-Lys(PpIX)]	62.3 ± 5.9
6	GnRH-I[^4^Lys(Bu), ^6^*D*-Lys(PpIX)]	89.8 ± 19.4
7	GnRH-II[^4^Lys(Bu), ^6^*D*-Lys(PpIX)]	71.1 ± 6.2
8	GnRH-III[^4^Lys(Bu), ^6^Asp(OMe), ^8^Lys(PpIX)]	81.8 ± 12.6
PpIX	Protoporphyrin IX	209.3 ± 119.5

**Table 4 ijms-20-05027-t004:** Patient characteristics at time of diagnosis.

Variable	No. of Patients
Total no. of patients	60
Sex	
Male	51
Female	9
Age (year)	
Mean	58.45 (41–77)
Localization	
Oropharynx	19
Larynx	24
Hypopharynx	16
Lingua	1
TNM ^1^ stage	
I	6
II	11
III	15
IV A	20
IV B	3
IV C	5

^1^ TNM: tumor, node, and metastasis, UICC TNM 7th edition.

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
