# Peer review of "Suitability of GnRH Receptors for Targeted Photodynamic Therapy in Head and Neck Cancers"

_ijms, 2019, doi:10.3390/ijms20205027_

Round 1
Reviewer 1 Report
The reviewed study concerns the synthesis and use of GnRH analogues conjugated with protoporphyrin IX for targeted phototherapy in the treatment of head and neck squamous cell carcinoma. The publication has a certain potential and its results may have a practical meaning, however, in its present form significant changes should be done.
Specific comments:
All the research should be more coherent, concentrated to Head and neck squamous cell carcinoma, authors many times describe other cancers that are not the subject of the study.
The result of quantitative expression of GnRH-R should be supplemented with more quantitative methods such as western blott (on the protein level) or QPCR (on mRNA).
Chapter of Peptide synthesis and GnRH-protoporphyrin IX conjugate synthesis is a methodical description and does not constitute a result. It should be transferred to the materials and methods section
In vitro studies must certainly be supported by in vivo experiments with the use of laboratory animals. The point is also question about the stability of the synthesized peptides in the blood.
The authors describe the expression of GnRH-R as membrane located, however, according to the reviewer they present a purely cytoplasmic reaction (concerns figure 5). Is the antibody specific? Please show positive and negative control reaction.
None of the diagrams shows statistical significance. What is the sample size in cell viability experience? The Kruscal Wallis statistical test with an appropriate post hoc test should be used, or ANOVA if the assumptions for carrying out this test are met. Please mark statistical significance on graphs.
The description of Cell cultures is not very detailed, e.g. lack of any information on cellular passages, medium changes, etc.
The MTT test should also be supplemented with apoptosis/ necrosis detection methods (TUNEL, flow cytometer).
Author Response
Reviewer 1
The reviewed study concerns the synthesis and use of GnRH analogues conjugated with protoporphyrin IX for targeted phototherapy in the treatment of head and neck squamous cell carcinoma. The publication has a certain potential and its results may have a practical meaning, however, in its present form significant changes should be done.
We thank the Reviewer for the thorough reading and evaluation of our manuscript and the valuable comments. Our detailed answers are given below.
Specific comments:
All the research should be more coherent, concentrated to Head and neck squamous cell carcinoma, authors many times describe other cancers that are not the subject of the study.
In this study we focused on the selection of a GnRH-based targeting moiety that can be used for targeted photodynamic therapy. GnRH derivatives as targeting moieties for drug delivery have mainly been used for other types of tumors, hence we had to mention them in the introduction. Their application in head and neck cancers, especially in photodynamic therapy is a novel aspect.
The result of quantitative expression of GnRH-R should be supplemented with more quantitative methods such as western blott (on the protein level) or QPCR (on mRNA).
With these experiments we only wanted to demonstrate that GnRH-R expression is significant in HNSCC, hereby the GnRH-based targeting of HNSCC may widely be used for targeted photodynamic therapy. However, these samples were not used for subsequent studies, therefore we do not think that further quantification would be important.
Moreover, the quantification of the GnRH receptors has already been done before on Detroit-562 cells (Murányi, J. et al., J. Pept. Sci. 2016, 22, 552-560.; reference 40 in the revised manuscript) that was applied in the latter experiments.
Chapter of Peptide synthesis and GnRH-protoporphyrin IX conjugate synthesis is a methodical description and does not constitute a result. It should be transferred to the materials and methods section
We agree with the Reviewer that the chapters “Peptide synthesis” and “GnRH-protoporphyrin IX conjugate synthesis” are too long and too specific for the results section, therefore we shortened them. However, we think that the synthesis of the novel peptides and conjugates belongs to the results, since from a chemical point of view, the synthesis of these compounds is not that trivial. Moreover, without the synthetic part, the subsequent in vitro studies could have not been done, which also shows the importance of the chemical sections. Hence, we would like to keep these shortened chapters in the results.
In vitro studies must certainly be supported by in vivo experiments with the use of laboratory animals. The point is also question about the stability of the synthesized peptides in the blood.
We thank the Reviewer for this comment. We are currently preparing the compounds in large scale for the in vivo studies, and our plan is to publish the new results in an independent research paper. Furthermore, the selection of the appropriate animal model for in vivo experiments is in progress, but this also takes time. Of course, we will do stability studies too, but we think that showing these results in this current paper would not provide added value. In addition, in case of photodynamic therapy not only the i.v. or i.p. injection might be the opportunity for the treatment, but also surface brushing with the solution of the conjugate before lightening might be an option. Therefore, a lot of things have to be studied before making appropriate in vivo studies.
Nevertheless, our earlier results show that similar (daunomycin as drug containing) GnRH conjugates have excellent serum stability, the intact bioconjugates are still detectable after 24 hours of incubation with human serum (Manea, M. et al., Bioconjugate Chem. 2011, 22, 1320-1329.; Orbán, E. et al., Amino acids 2011, 41, 469-483.; Schuster, S. et al., Beilstein J. Org. Chem. 2018, 14, 756-771.). We presume that PpIX as drug molecule would not drastically change the stability of our conjugates compared to the daunomycin containing ones.
The authors describe the expression of GnRH-R as membrane located, however, according to the reviewer they present a purely cytoplasmic reaction (concerns figure 5). Is the antibody specific? Please show positive and negative control reaction.
The antibody has already been used in our earlier publication (Murányi, J. et al., J. Pept. Sci. 2016, 22, 552-560.; reference 40 in the revised manuscript). Here, we confirmed that the chosen antibody is specific, showing positive and negative controls as well.
We performed the confocal laser scanning microscopy measurements more carefully again, and we provide a revised figure that proves that the GnRH-R is found in the membrane in high concentrations.
None of the diagrams shows statistical significance. What is the sample size in cell viability experience? The Kruscal Wallis statistical test with an appropriate post hoc test should be used, or ANOVA if the assumptions for carrying out this test are met. Please mark statistical significance on graphs.
6000 cells/well were seeded on 96-well plates 48 hours prior to the treatment, and each point was measured using 3 parallels and each experiment was repeated twice, as described in the Methods part.
According to the Reviewer’s recommendation, we performed the statistical evaluation of the data by using the multiple t-test algorithm of GraphPad Prism 8.2.0 software. Statistical significance was determined using the Holm-Sidak method and was marked on the revised graphs 6 and 7, where the significance levels correspond to *p < 0.05; **p < 0.01; ***p < 0.001.
The description of Cell cultures is not very detailed, e.g. lack of any information on cellular passages, medium changes, etc.
We provide a more detailed description for the cell cultures in the Methods section: “For the experiments, Detroit-562 cells with passage number 12 were thawed and serially passaged until passage 35. For new passages, cells were seeded into T25 flasks and grown to 80–90% confluence. Then the cells were rinsed with phosphate buffered saline and removed by adding Trypsin–EDTA. The cell number was established using hemocytometer.”
The MTT test should also be supplemented with apoptosis/ necrosis detection methods (TUNEL, flow cytometer).
We thank the Reviewer for this valuable idea that could definitely provide more information about the mechanism of activity of the prepared conjugates. We also thought about performing similar assays, but at the moment we do not have the technical background for such measurements. Moreover, we are planning to do other studies of mechanism of action, too (such as ROS experiments). All would be part of an independent publication.
Reviewer 2 Report
Review Report-IJMS 570410
Suitability of GnRH receptors for targeted photodynamic therapy in head and neck cancers
Authors have investigated the therapeutic relevance of Gonadotropin releasing hormone-receptor targeting in head and neck cancers by evaluating the newly synthesized conjugates on Detroit-562 cell line. The paper is of interest and the content fits well under the scope of this journal. Manuscript can be accepted after minor revisions.
N- and C- terminal parts (line 60-61) represent which compound? Mention it clearly in Introduction section. Introduction: Give a brief background of PpIX in the Introduction (why this was chosen? Therapeutic applications etc.) Cut down the extensive information on GnRH-I, II, III analogues’ previous research. Emphasize the rationale of the study at the end. Figure 3: Name the starting compounds and the end product in the figure. Explain the reaction conditions in the caption. Section 2.4, Table 2: Authors have presented the binding potency results in terms of IC50 values. This seems to be irrelevant. Please present dissociation constant and maximal binding capacity values (Kd and Bmax). Also, Include unconjugated PpIX in the experiment as a negative control. Write the procedure in detail in section 4.9 along with the previous paper citation. Section 2.7: In the optimization study, why the effect of different irradiation times was not studied with respect to PpIX? Section 2.8: Along with cell viability inhibition data, present IC50 values for each compound in the results section.Author Response
Reviewer 2
Suitability of GnRH receptors for targeted photodynamic therapy in head and neck cancers
Authors have investigated the therapeutic relevance of Gonadotropin releasing hormone-receptor targeting in head and neck cancers by evaluating the newly synthesized conjugates on Detroit-562 cell line. The paper is of interest and the content fits well under the scope of this journal. Manuscript can be accepted after minor revisions.
We thank the Reviewer for the thorough reading and positive evaluation of our manuscript and the valuable comments. Our detailed answers are given below.
N- and C- terminal parts (line 60-61) represent which compound? Mention it clearly in Introduction section.
According to the Reviewer’s comment, we rephrased that sentence to “The N- and C-terminal parts of GnRH (amino acids 1-4 and 9-10),…”
Introduction: Give a brief background of PpIX in the Introduction (why this was chosen? Therapeutic applications etc.)
According to the Reviewer’s recommendation, we provide a short description about PpIX in the Introduction part: “Protoporphyrin IX (PpIX) is an endogenous photosensitizer, it is the last intermediate in heme biosynthesis. Endogenous PpIX-based strategies have been approved by the FDA for treating cancer, where δ-aminolevulinic acid (ALA, the first intermediate in heme biosynthesis) is administered orally or locally to generate PpIX biosynthesis. Unfortunately, the generated PpIX does not only accumulate in cancer cells but also in healthy cells, such as the marrow, the circulating erythrocytes and the liver, causing photosensitivity or liver damage [32]. PpIX has two carboxyl groups that are suitable for the conjugation of a targeting moiety giving the opportunity to increase the selectivity. Hence recently, PpIX has also been studied as exogenous photosensitizer conjugated to peptides [25, 33], nanoparticles [34-36] or quantum dots [37] and encapsulated into polymer dendrimers [38-39].”
In this study we focused on the selection of appropriate targeting moieties for GnRH-based targeted photodynamic therapy. For this purpose, we chose a fairly cheap, but good photosensitizer that is often used in the published papers. We know that there are much better (and more expensive) compounds for photodynamic therapy, and we would like to use one of them only with the best carrier. This novel conjugate may also be applied for the in vivo tests.
Cut down the extensive information on GnRH-I, II, III analogues’ previous research. Emphasize the rationale of the study at the end.
Several GnRH-I, II and III derivatives as targeting moieties have already been used for drug delivery. According to our opinion, we only provided a very short overview for all analogs that might be useful in this special issue, furthermore, the structural changes that were used to design the novel peptides and bioconjugates were also mainly known, hence we had to mention all of them in the introduction, as well.
Figure 3: Name the starting compounds and the end product in the figure. Explain the reaction conditions in the caption.
We thank the Reviewer to highlight this inadequacy, we changed Figure 3 and its caption in section 2.3.
Section 2.4, Table 2: Authors have presented the binding potency results in terms of IC50 values. This seems to be irrelevant. Please present dissociation constant and maximal binding capacity values (Kd and Bmax). Also, Include unconjugated PpIX in the experiment as a negative control. Write the procedure in detail in section 4.9 along with the previous paper citation.
We thank the Reviewer for the valuable suggestions. In the present study we did not intend to demonstrate the presence of specific GnRH binding sites and characteristics of binding of [125I]-[6D-Trp]-GnRH-I as radioligand to membrane homogenates of prostate cancer specimens and human pituitary samples since it was published earlier. In our study we investigated the binding potency of our newly synthesized compounds to human pituitary and prostate cancer samples that express specific, high affinity GnRH receptors in relatively high concentration. It was carried out by displacement analysis type of ligand competition assay that investigates equilibrium binding at a fixed concentration of radioligand (125I-radiolabelled GnRH-I agonist triptorelin ([125I]-[6D-Trp]-GnRH-I)) in the presence of increasing concentrations of the unlabeled peptides/conjugates. The displacement efficacy, and thereby the binding potency can be characterized by a concentration where half of the receptor bound 125I-radiolabelled GnRH-I agonist triptorelin is displaced by the competitor compound (IC50 value). Based on our best knowledge and according to the scientific literature, this type of binding assay is appropriate, and such in vitro experimental method can be properly used for the determination of binding potency of newly synthesized compounds.
However, in order to answer the question, we have collected Kd and Bmax values of [125I]-[6D-Trp]-GnRH-I binding. Analyses of the displacement of the radiotracer [125I]-[6D-Trp]-GnRH-I by the same unlabeled peptide triptorelin revealed that GnRH binding sites expressed in human prostate cancer samples had a Kd values of 1.6 – 4.7 nM with a Bmax of 415.3 – 482. 2 fmol/mg protein.
In addition, based on the request of the Reviewer, a new sentence was inserted into the manuscript in line 161: „At a concentration of 10-6 M unconjugated PpIX and several other peptides structurally related or structurally and/or functionally unrelated to GnRH, such as somatostatin-14, hGHRH, EGF, IGF-I, glucagon and VIP did not inhibit the binding of [125I]-[6D-Trp]-GnRH-I.”
In view of the request of the Reviewer, the original text in section 4.9 was revised to give more details about the method we used. „... The binding affinities of the GnRH-analogs and GnRH-protoporphyrin conjugates to GnRH-R-I were determined by displacement of [125I]-[6D-Trp]-GnRH-I performing an in vitro ligand competition assay. The assay was performed as described earlier [13-15, 30, 34,43]. In brief, membrane homogenates containing 50-160 µg protein were incubated in duplicate with 60-80.000 cpm [125I]-[6D-Trp]-GnRH-I in the presence of increasing concentrations (10-12 - 10-6 M) of nonradioactive unlabeled peptides/conjugates as competitors in a total volume of 150 µl of binding buffer. At the end of the incubation, 125 µl aliquots of suspension were transferred onto the top of 1 ml of ice-cold binding buffer containing 1.5 % BSA in microcentrifuge tubes. The tubes were centrifuged at 12.000x g for 3 min at 4 ºC. Supernatants were aspirated and the pellet was counted in a gamma counter. Protein concentration was determined by the method of Bradford using a Bio-Rad protein assay kit (Bio-Rad Laboratories, USA). The LIGAND-PC computerized curve-fitting program of Munson and Rodbard was used to determine the receptor binding characteristics and IC50 values."
Section 2.7: In the optimization study, why the effect of different irradiation times was not studied with respect to PpIX?
Our goal was to design the best targeting moiety for the targeted photodynamic therapy, which can also help the cellular uptake of the PpIX conjugate. Therefore, we selected only one of the conjugates to optimize the irradiation time and the attainable effect, since this could provide the data we needed for the next steps. Then, we chose only the best irradiation time to compare all the conjugates and PpIX. From the data observed, we could conclude that the peptides could help with the uptake of the photosensitizer.
In case of the optimization of the incubation time, we wanted to prove that the peptide-PpIX conjugates enter the cells by receptor mediated endocytosis (the uptake and the effect increase by the time), while PpIX enters the cells by diffusion showing less difference with increased incubation time.
Section 2.8: Along with cell viability inhibition data, present IC50 values for each compound in the results section.
As the Reviewer required, we present a table of the IC50 values for all the peptide-PpIX conjugates and the pure PpIX in section 2.8.